# Effect of the Production Parameters and In Vitro Digestion on the Content of Polyphenolic Compounds, Phenolic Acids, and Antiradical Properties of Innovative Snacks Enriched with Wild Garlic (*Allium ursinum* L.) Leaves

**DOI:** 10.3390/ijms232214458

**Published:** 2022-11-21

**Authors:** Kamila Kasprzak-Drozd, Tomasz Oniszczuk, Iwona Kowalska, Jarosław Mołdoch, Maciej Combrzyński, Marek Gancarz, Bohdan Dobrzański, Adrianna Kondracka, Anna Oniszczuk

**Affiliations:** 1Department of Inorganic Chemistry, Medical University of Lublin, Chodźki 4a, 20-093 Lublin, Poland; 2Department of Thermal Technology and Food Process Engineering, University of Life Sciences in Lublin, Głęboka 31, 20-612 Lublin, Poland; 3Department of Biochemistry and Crop Quality, Institute of Soil Science and Plant Cultivation, State Research Institute, 24-100 Puławy, Poland; 4Faculty of Production and Power Engineering, University of Agriculture in Kraków, Balicka 116B, 30-149 Kraków, Poland; 5Institute of Agrophysics, Polish Academy of Sciences, Doświadczalna 4, 20-290 Lublin, Poland; 6Pomology, Nursery and Enology Department, University of Life Sciences in Lublin, Głęboka 28, 20-400 Lublin, Poland; 7Department of Obstetrics and Pathology of Pregnancy, Medical University of Lublin, Staszica 16, 20-081 Lublin, Poland

**Keywords:** dietary polyphenols, high performance liquid chromatography, functional food, in vitro two-stage simulated gastrointestinal digestion, wild garlic, antioxidant activity, extrusion-cooking, processing parameters

## Abstract

A new type of corn snack has been created containing additions of wild garlic (*Allium ursinum* L.). This medicinal and dietary plant has a long tradition of use in folk medicine. However, studies on wild garlic composition and activity are fairly recent and scarce. This research aimed to investigate the influence of the screw speed and *A. ursinum* amounts on the antiradical properties as well as the content of polyphenolic compounds and individual phenolic acids of innovative snacks enriched with wild garlic leaves. The highest radical scavenging activity and content of polyphenols and phenolic acids were found in the snacks enriched with 4% wild garlic produced using screw speed 120 rpm. The obtained findings demonstrated that snacks enriched with wild garlic are a rich source of polyphenolic compounds. Since the concentration of such compounds is affected by many factors, e.g., plant material, presence of other compounds, and digestion, the second aim of this study was to determine radical scavenging activity, the content of polyphenols, and individual phenolic acids of snacks after in vitro simulated gastrointestinal digestion. Using an in vitro two-stage model, authors noted a significant difference between the concentration of polyphenolic compounds and the polyphenol content of the plant material before digestion.

## 1. Introduction

Polyphenolic compounds, of which more than 8000 chemical structures have been described, are one of the most diverse and widely distributed groups of secondary plant metabolites. They are an essential component of a balanced human diet [1]. The common feature of polyphenols is the presence of at least two hydroxyl groups attached to one or more aromatic rings [2,3]. The most adopted classification places the phenolics within one of two groups: flavonoids (e.g., flavanols, flavanones, and anthocyanins) and non-flavonoids (e.g., phenolic acids, stilbenes, lignans) polyphenols. Phenolic acids are further subdivided into derivatives of benzoic and cinnamic acid in terms of their chemical structure [1].

Phenolic compounds are direct antioxidants but show indirect antioxidant activity by inducing endogenous protective enzymes and positive regulatory action with regard to the signaling pathways [4,5]. The antioxidant activity of polyphenols is determined by the presence of hydroxyl groups. The strength of this action depends on their number and position in the ring structure, esterification, or proximity to other substituents [3,6]. The predominant mechanism of the antioxidant activity of these compounds is believed to be radical scavenging via hydrogen atom donation, although other types of mechanisms are recognized [3,4].

Oxidative stress is an imbalance state between ROS (reactive oxygen species) and antioxidant defenses. It can lead to various pathological conditions in the body, such as tissue injury and accelerated cellular death. The potential of plant products to serve as antioxidants to protect against ROS and various diseases induced by free radicals has been explored [7]. Scavenging free radicals through the action of compounds with antioxidant properties (such as polyphenols) reduces and prevents damage caused by oxidative stress [8]. Due to the fact that polyphenols exhibit a number of properties that are beneficial to health, in this case, many times representing a specific therapeutic potential, they are a frequent topic of research work. In most plant materials, they occur as compounds accompanying other active substances. Therefore the possible synergistic effect of their action must be taken into account [6]. The DPPH (2,2-diphenyl-1-picrylhydrazyl radical) assay is used to determine the radical-scavenging activity of samples [7,9].

In addition to the antioxidant activity, the therapeutic effects of polyphenols have been proven: e.g., anti-inflammatory and analgesic activity [10], as inhibitors of neurodegenerative processes [6], dyslipidemia and cardiovascular disease [11], in treating overweight and obesity [12]. Of note, most naturally occurring ingredients with consistently reported anticancer efficacy contain high levels of polyphenols [1]. 

A plant rich in phenolic compounds is ramson (wild garlic or bear’s garlic, *Allium ursinum* L.), a dietary and medicinal plant with a long tradition of use in folk medicine. However, studies on its composition and pharmacological activity are fairly recent and scarce [13]. The species name “ursinum” is of Latin origin, derived from “ursus” (bear). It is related to folk stories, according to which bears, after awakening from winter hibernation, consume this plant to regain strength [14].

*Allium ursinum* L. has a distinct garlic-like scent associated with the presence of sulfur-containing compounds. These compounds are undoubtedly the most important constituents of ramson, both in terms of chemotaxonomic value and pharmacological activity. Of the various sulfur compounds present in this plant, glutamyl peptides and sulfoxides are considered primary. Apart from sulfur-containing substances, *A. ursinum* has also been reported to be a good source of phenolic compounds. The leaves contain free and bound forms of phenolic acids (protokatechuic, 4-OH-benzoic, vanillic, caffeic, syringic, coumaric, ferulic, and sinapic). As far as flavonoids, the leaves of ramson are abundant, predominantly in kaempferol derivatives [15].

The addition of wild garlic leaves to food allows for obtaining products with an extraordinary concentration of substances promoting health. The benefits of such additives include increasing the amount of dietary fiber-containing polyphenolic compounds of the nature of antioxidants, as well as enhancing the levels of essential oils and sulfur compounds.

Ramson has a much milder effect on the gastrointestinal tract than in the case of the addition of common garlic (*Allium sativum* L.). Common garlic, unlike wild garlic, is characterized by a more pronounced spicy taste and aroma, and it also causes an unpleasant smell from the mouth. The latter effect—the sulfur smell of the breath after eating even small amounts of garlic—is the result of the presence of bactericidal, antiviral, and antifungal allicin, the metabolites of which are excreted through the lungs. This organosulfur chemical can also damage the digestive tract if ingested in large amounts. The literature data [15,16] shows that wild garlic contains more than two times less of this compound in favor of three times more content of other sulfur-organic compounds that are equally beneficial for health but less burdensome to the gastrointestinal tract. The presence of antioxidant polyphenolic compounds in the plant—including phenolic acids—protects the body against the harmful effects of free radicals, the excess of which in the body can cause neurodegenerative diseases, cancer, or cardiovascular diseases. Due to its pro-health properties, high content of polyphenols, and the fact that no one has used ‘bear’s garlic’ in the production of functional food so far, authors decided to use this plant as an addition to snacks.

In order to exhaustively assess the potential biological properties of polyphenolic compounds that naturally occurred in plants, it is essential to study their bioaccessibility, bioavailability, and release efficiency, as undegraded bioactive compounds during digestion (bioaccessibility) can be available for absorption, especially in the intestine and promote biological action (bioavailability). In addition, it is important to study their synergistic action and coexistence in the plant matrix. In vitro models of the gastrointestinal tract are used for this purpose [8,17]. The complex biotransformation of plant secondary metabolites after ingestion and their low bioavailability can lead to tissue concentrations well below those exhibiting antioxidant capacity in vitro. Importantly, inside the gastrointestinal tract, the antioxidant potential of dietary components may play a key role in the body’s defense at the systemic level [18].

Factors in the bioaccessibility of polyphenols include their release from the food matrix, particle size, their hydrophilic/lipophilic balance as related to their glycosylation, different pH-dependent transformations (degradation, hydrolysis, and oxidation within the gastrointestinal tract), and also interactions between polyphenols and food components. The absorption of some the free polyphenols (e.g., phenolic acids) occurs in the stomach, and these compounds can conjugate with glucuronic acid. The contribution of the intestinal step to the bioaccessibility of polyphenols depends on many parameters. For example, impact of intestinal enzymes on the residual matrix could increase the level of phenolic compounds. Absorption of aglycones and their glucosylated forms by the small intestine the process can take place in two ways: by passive diffusion or active transport. Undigested polyphenols pass into the large intestine, where they are further degraded by colonic microflora. Finally, metabolites of all these compounds lead to benzoic acid generation [17].

This study was primarily aimed at investigating the influence of the production parameters (screw rotational speed and *A. ursinum* addition) on the content of polyphenolic compounds, individual phenolic acids, and antiradical properties of innovative snacks enriched with wild garlic leaves.

Before bioactive compounds can be used by human organisms, they must be processed in the digestive tract in order to be assimilated in the proper form. Therefore, studies that are based only on the determination of the concentration of phytochemicals in plant material do not give proper information about the bioavailable level of these compounds. In contrast, analyses using an in vitro digestion system provide more detailed data on the concentration of test components after ingestion and digestion in the gastrointestinal tract in vivo [17]. Therefore, the secondary aim of this study was to determine the content of polyphenolic compounds, individual phenolic acids, and antiradical properties of the snacks after in vitro simulated gastrointestinal digestion.

## 2. Results and Discussion

### 2.1. Influence of A. ursinum Addition and Screw Speed on Polyphenols Content, Free Phenolic Acid Content and Antioxidant Properties of Snacks

The wide spectrum of biological activities obtained from wild garlic and the presence of chemical compounds with high therapeutic possibilities make this plant a candidate for the development of functional food and food supplements. Here, the used extraction method can significantly affect the quality and concentration of the targeted compounds. Extraction of active compounds from wild garlic using an ultrasound-assisted (UAE) process was recommended by Tomšik et al. [16] as an efficient, inexpensive, and simple existing extraction system that could be suitably upscaled for largescale preparations. UAE at elevated temperatures with 80% aqueous ethanol as an extractant was also previously used by authors [19,20] to obtain polyphenol extracts from snacks enriched with herbs. As UAE was demonstrated to be the optimum technique for isolating phenolic acids from functional foods, the authors decided to use it in this experiment.

The samples—corn snacks with different quantities of wild garlic leaves (0, 1, 2, 3, and 4%) were produced using extrusion cooking at various screw speeds. In the first phase of the experiment, the authors examined the total content of polyphenolic compounds. Polyphenols are secondary metabolites of plants, present in the human diet, and used for medicinal and cosmetic purposes. They have strong antioxidant, anti-radical and pharmacological properties. The results showed that their content had increased significantly due to wild garlic addition (Table 1), notably with increased screw speed. The highest total content of polyphenols (as per gallic acid equivalents; GAE) was reported in snacks with a 4% addition of the *A. ursinum*, while the lowest was seen in a sample without such functional additive.

*A. ursinum* leaves were previously reported by Djurdjevic et al. [21] to contain free forms of ferulic and vanillic acids and bound forms of coumaric, ferulic, and vanillic acids. Phenolic acids are present in plant, most often as glycosides and esters. Acid (or base) hydrolysis transforms glycosylated and esterified phenolics into their aglycones and is often used during the analysis of phenolic acids in plant material [22]. For this reason, in the first step of the experiment, the content of polyphenols and free phenolic acids in the product was determined after acid hydrolysis. The phenolic acid content in extracts was assessed by applying reversed-phase ultra-high pressure liquid chromatography using a photodiode array detector coupled to a triple-quadrupole mass spectrometer. The following phenolic acids were identified in the samples: protocatechuic, 4-OH-benzoic, vanillic, syringic, salicylic (benzoic acid derivatives), and caffeic, coumaric, ferulic, sinapic (cinnamic acid derivatives) (Table 2, Figure 1 and Figure 2).

Ferulic acid was the prevailing phenolic acid. These results are consistent with those obtained by Djurdjevic et al. [21], who identified three phenolic acids in wild garlic extracts (*p*-coumaric, ferulic and vanillic), and Pop et al. [23], who quantified coumaric, ferulic and sinapic acids. However, in this study, the authors determined five additional phenolic acids in all snacks enriched with wild garlic.

As in the analysis of total polyphenols, also in this research, the content of active compounds in general increased as wild garlic leaves were added to the snacks. The comparatively proportional increase in the content of phenolic acids demonstrates extrusion process, under conditions of high temperature and high pressure, did not degrade the active compounds present in snacks enhanced with this raw material. In this regard, slight deviations may indicate that the ingredients had not been mixed properly during the preparation or production stage. 

*A. ursinum* is the source of many antioxidant molecules [23]. Therefore the quantity of total phenolic compounds (generally assumed to be responsible for the antioxidant activity of plant extracts) [23] was ascertained for snacks with the addition of wild garlic (Table 3). In the next phase of the experiment, the authors determined the DPPH free radical scavenging potential of the tested samples using a UV-VIS spectrophotometer. The obtained results showed that the antioxidant activity of snacks increased with the amount of added *A. ursinum*. The highest DPPH free radical scavenging activity was seen in the products supplemented with 4% of wild garlic and at 120 rpm. The maximum free radical scavenging ability by all extracts was obtained after 15 min. The results obtained by Pop et al. [23] by means of applying the DPPH radical-scavenging test, showed that wild garlic represents important sources of bioactive phenolic acids, volatile and sulfur compounds, and flavonoids) with strong in vitro antioxidant activity.

Previous research have shown that aglycones have a higher antioxidant activity than their glycosidic compounds or are connected by other types of bonds [24]. Sani et al. [25] studied the influence of acidic hydrolysis on the yield, total phenolic content, and antioxidative capacity of methanolic extract of germinated brown rice. These authors applied total phenolic content and DPPH radical scavenging for the measurement of antioxidant ability. In the study presented by Sani et al. [25], there were a significant difference in the total phenolic content and DPPH radical scavenging assay results when comparing neutral with acidic hydrolysis. Snack samples were further tested using high performance liquid chromatography to determine the individual phenolic compound levels in different hydrolytic media contributing to the antioxidant effects. This study revealed that acidic hydrolysis could improve the polyphenols content and antioxidant properties of germinated brown rice.

Furthermore, research has shown that the antioxidant activity of polyphenolic compounds is dependent on the number of hydroxyl groups in the molecules and can be changed by spherical effects, as well as by interactions of the compounds existing in the matrix and in the extracts [26]. In addition, it has been examined that the antioxidant activity of products is dependent on their composition. Korus et al. [27] found a lower antioxidant potential for red beans compared to black-brown and cream beans, even though dark-red bean extrudates demonstrated higher phenolic content compared to black-brown and creamy bean products. In the case of these innovative snacks enriched with *A. ursinum*, the product radical-scavenging activity measured after 15 min was positively correlated with the total content of polyphenols and free phenolic acids (Table 4). Very high positive correlations were established between the wild garlic leaves addition and radical-scavenging activity for both screw speeds.

Polyphenols content can vary due to food manufacturing parameters [28]. Multari et al. [24] state that high-temperature treatment can improve the release of phenolic compounds bound to the cell wall structures. Furthermore, other compounds that have a beneficial effect on the human body can appear.

Extrusion-cooking is the short-time method of processing starchy raw materials under high-temperature (120–200 °C) and high pressure (20 MPa) conditions. The intense processing of mechanical shearing results in a deep transformation of individual components. Temperature, screw speed, moisture content, and residence time distribution during the process are crucial for the polyphenols content antioxidant activity of the product [29] and antinutritional factors level [30]. Khanal et al. [31] showed the effects of extrusion-cooking on procyanidin in grape seed and pomace. These authors have demonstrated that this method increases the levels of low-molecular-weight bioactive compounds (e.g., procyanidin) and releases biologically important monomers and dimers from polymer chains [31]. The intensity of changes therein depends on the properties of the raw material, e.g., humidity and the processing parameters. A high homogenization of ingredients leads to a decrease in diffusion barriers and the breaking down of chemical bonds. This factor results in the heightened reactivity of the components. Optimized extrusion-cooking conditions may therefore, release phenolic acids from the chemical bonds that they create with other compounds without deactivating aglycones [29,30,31]. This is due to the cracking of the rigid cell walls and other plant cell components. The breakdown of the glycosidic bonds, which leads to the formation of aglycons, is also conducive to an increased antiradical capacity of products [32]. A Diagram of snack production using extrusion cooking is presented in Figure 3.

The next stage of the research focused on the influence of one extrusion-cooking condition—the screw rotational speed, on the total content of polyphenols and free phenolic acids, as well as the antioxidant properties of extruded snacks. The obtained findings demonstrated that all snacks enriched with the wild garlic screw speed of 120 rpm resulted in a higher content of polyphenols and free phenolic acids and a higher ability to scavenge DPPH than the speed of 80 rpm (Table 1, Table 2 and Table 3). Alonso et al. [33] examined that the most important factors stimulating the transformation of raw material during the extrusion-cooking process are high temperatures and mechanical factors related to shear forces that increase along with the increase in screw rotational speed. It is possible that the extrusion conditions at 80 rpm are too mild, and they do not enable the release of some phenolics from the processed raw material. The rotation speed of 80 rpm was found to give higher contents than 120 rpm for only two phenolic acids. They were salicylic and sinapic acids, which have lower decomposition temperatures than most of the analyzed acids. So, it is possible that for these compounds, 120 rpm induced too drastic production conditions and brought about their decomposition.

Many previous studies have revealed antioxidant activity increase in extruded products with an increased temperature of processing [33,34,35]. This effect is explained due to the presence of products formed during Maillard reactions—which can have high radical scavenging activity.

Currently popular in food technology applications, the extrusion-cooking process can provide diversified properties within the final product. The effects of temperature, pressure, and shear forces on moist raw material induce profound changes in a very short time. They include the inactivation of anti-nutritive factors, enhanced digestibility of nutrients, and modified sensory characteristics. The type and intensification of these changes depend on the parameters of the extrusion-cooking procedure (e.g., extruder screw rotational speed, temperature) as well as on the properties of the raw materials. The high degree of component mixing reduces diffusion barriers and breaks chemical bonds. This results in increased reactivity of the ingredients [36]. For this reason, the proper selection of parameters is crucial for the production process.

### 2.2. The Digestability of the Snack’s Polyphenols Using Two-Stage In Vitro Human Digestion Model

Credible reports on the breakdown of food ingredients in the stomach are also crucial for assessing phytochemical bioaccessibility. Gastric digestion is multiple stages that includes mechanical actions and the effect of gastric fluids. Gastric juice contains hydrochloric acid, lipase, pepsinogens, mucus, electrolytes, and water. Hydrochloric acid supports the denaturation of proteins, and it activates pepsin. The duration of this phase lasts 2 to 4 h. The gastric pH in healthy human subjects in the fasted state is in the range of 1.3 to 2.5; the intake of a meal generally increases the pH to above 4.5. Unfortunately, most in vitro models include a pH below 2.5, which is a pH related to the human fasting state more than to real food digestion. The change in gastric pH is taken into consideration only in dynamic models [37,38]. The absorption of free phenolic acids occurs in the stomach, and phenolic acids can conjugate with glucuronic acid. The in vitro small intestinal digestion model simulates the time, temperature, pH, and composition of pancreatic juice, including electrolytes, bile salts, and enzymes. In the fed state, pH can be from 5.4–7.5 in the duodenum [39], to 5.3–8.1 in the jejunum, and up to 7.0–7.5 in the ileum. The contribution of the intestinal step to the bioaccessibility of polyphenols depends on many parameters. The effect of intestinal enzymes on the residual matrix could, for example, increase the concentration of phenolic compounds. Moreover, degradation or isomerization of these compounds can occur due to catalyzation by the presence of oxygen and/or transition metal ions. In addition, specific absorption of aglycones and their glucosylated forms by the small intestine can come about by passive diffusion or active transport [40].

In the human body, esterified polyphenols are degraded in the large intestine by microbial esterases. Undigested polyphenols pass into the large intestine, where they are further degraded by colonic microflora. Depending on the polyphenol’s structure, a large variety of compounds can be formed. Finally, metabolites of all these compounds lead to benzoic acid generation [41].

In this study, a two-stage in vitro digestion model, including gastric and duodenal phases, was used. Before carrying out in vitro digestion, authors examined the total content of polyphenolic components, the content of free phenolic acids, and the antioxidant properties of the selected samples (Table 5, snacks enriched with 0%, 2%, and 4% of wild garlic extruded at 120 rpm) without prior hydrolysis, hence the much lower content of free phenolic acids before digestion (Table 6) in relation to its content after hydrolysis (Table 2). For all investigated snacks, the concentration of polyphenols after in vitro digestion was significantly reduced, both after gastric and duodenal digestion, compared with the samples before digestion (Table 5). The content of free phenolic acids also decreased drastically after the first stage of in vitro digestion (gastric), which was deepened during the second stage (duodenal). Acids such as protokatechuic, 4-OH-benzoic, vanillic, syringic, sinapic, and salicylic have not been found in the mixture after the first phase of digestion (Table 6).

There are reports in which the total concentration of polyphenols was reduced during the gastric digestion stage, as in this study. This tendency has been observed for aqueous infusions from *Capparis spinosa* L., *Crithmum maritimum* L. [42], chamomile tea, yerba mate, coffee-like substitutes, and coffee blend. The decrease in active compounds content was further exacerbated in the subsequent intestinal part of the in vitro digestion [43]. Similarly, Dacrema et al. [44] observed a drop in the content of individual polyphenolic compounds after in vitro digestion of fireweed extract. These Authors reported a loss of individual polyphenolic compounds after the orogastric phase (in the range of 1.92–84.17%) and a decrease (11.83–98.07%) after the duodenal phase.

In plant-processing by-products of black carrot, a decrease in phenolic acids content (chlorogenic, neochlorogenic, and cryptochlorogenic) was also observed during the gastric stage of in vitro digestion, which escalated at further stages of digestion. Contrastingly, ferulic and caffeic acid mostly demonstrated an increase in bioavailability as compared to undigested samples [45]. As in the author’s work, Majdoub et al. [46] have also noted a decrease in certain compounds post-in vitro gastric and gastric + duodenal digestion. In particular, caffeoylquinic acids were found to be partially decreased after 20 min of gastric digestion. Likewise, coumaroylquinic acids only minutely persisted throughout the simulated human digestion as a result of the degradation occurring in the gastric and duodenal compartments. However, quinic acid was only found in the samples obtained during duodenal digestion. This fact suggests that this compound was derived from the degradation of the more complex coumaroylquinic and caffeoylquinic acids. Moreover, the work of Majdoub et al. [46] saw that hibiscus acid was present in the undigested extract and persisted, albeit in lower concentrations, throughout the simulated digestion. Other authors have indicated high polyphenol stability in the gastric phase and their degradation at the intestinal level. Gayoso et al. [47] examined the effect of in vitro gastrointestinal digestion methods using three static models on the stability and bioaccessibility of phenolic compounds. When the results were referred to mg/mg lyophilized digested sample, the remaining % of the sample decreased to 67% and 68% (oral and gastric, respectively) for rosmarinic acid, and 75% and 78% (oral and gastric, respectively) in the case of caffeic acid. No remarkable differences were observed for rutin between the initial level subjected to digestion and the levels recovered during the oral and gastric steps.

It is difficult to compare bioaccessibility studies due to the many variables that may influence gastrointestinal digestion. Differences in results can be dictated by the effect of the plant/food matrix, the heterogeneity of analyzed plant materials, their degree of processing, as well as the in vitro digestion methodology. Pure compounds also show high variability. For example, for rutin, the % of loss after intestinal digestion was found to be from only 3% to total loss. In the case of quercetin, the results ranged from 5.8% [48] to total loss [42], and for chlorogenic acid, from 44% to 95.7% [42]. It is certain that the digestion methodology is a key factor for assessing the bioaccessibility of polyphenols. 

This research has shown the in vitro digestion procedure reduced the antioxidant activity of snacks (Table 7). The decrease in antioxidant activity is consistent with the results presented by other authors. The study of the effect of in vitro digestion on the antioxidant activity of phenolic demonstrated a decrease in activity for rosmarinic acid (24–36%), caffeic acid (12–19%) and no change in antioxidant capacity for rutin [47]. A similar tendency of the reduction in antioxidant potential for plant-processing by-products of black carrot was observed during the gastric stage of in vitro digestion. At the further phases of digestion, the results were divergent, indicating an increase or decrease depending on the method and raw material [45]. The reduction in antioxidant activity during the intestinal part of in vitro digestion was observed in this study and in the work of other authors. This fact can be explained by the structural reorganization of some compounds during a change in pH to slightly alkaline.

pH conditions have a significant influence on both in vitro and in vivo studies concerning the antioxidant activity of polyphenols. It is significantly different in acidic and alkaline environments in relation to neutral environments. This fact is crucial for the theoretical deliberation of the influence of the pH of individual parts of the digestive tract on the structures and activity of plant metabolites. However, available study results are limited. It is known that the antioxidant activity of extracts depends on the number of OH groups in their main compounds, as well as their hydrogen-donating abilities. Moreover, additional OH groups in *ortho*-positions have a positive effect on the antioxidant activity of phenolic compounds, especially at pH 4. Thus, in order to analyze the pH influence on the activity of food polyphenols, each component must be considered [49].

Lettuce (*L. sativa*) extract is rich in polyphenols such as caffeic acid derivatives, chlorogenic acids, and flavonoids [50]. The results of a study involving lettuce extract showed that free radical scavenging potential increases with increasing pH [51]. Other results were obtained for sweet potato leaf extract [52]. This material is rich in chlorogenic and caffeic acid derivatives. In this case, a slightly alkaline (pH 8) environment had a negative influence on antioxidant ability, while neutral and weak acidic environments gave an increase in activity. Why regard to this effect, interesting experiments were performed for honey. The analysis confirmed that the substance is rich in flavonoids (quercetin, luteolin, hesperitin, and apigenin) and phenolic acids such as cinnamic, benzoic, vanillic, coumaric, caffeic, chlorogenic and ellagic, and the authors observed significantly decreasing activity with increasing pH.

These discrepancies can result from different species of analyzed plant raw materials, their distinct characteristics, chemical structure and initial concentration of bioactive compounds, the methodology for the determination of antioxidant potential, and the in vitro digestion procedure [37].

Phenolic compounds during gastrointestinal digestion can be hydrolyzed as a consequence of the acid environment of the stomach, the alkaline environment of the intestine, and the action of digestive enzymes [53]. Furthermore, these compounds have the ability to bind with other ingredients of the food matrix, resulting in the formation of complexes that may also contribute to the reduction in their antioxidant activity.

## 3. Materials and Methods

### 3.1. Plant Materials

*Allium ursinum* L. leaves (series nr 5902741007841) were purchased from “Dary Natury Sp z o.o.” herbal industrial (Grodzisk, Poland). Before the extraction, the dry plant material was milled and sieved. Corn grits were purchased at the local market (distributor Vegetus, Poland).

### 3.2. Extrusion-Cooking Procedure

Blends of corn grits and ground wild garlic leaves were prepared by mixing dry components in the ratios of 100:0, 99:1, 98:2, 97:3, and 96:4 on a weight basis. The blended samples were conditioned to 15% of moisture content by spraying with a calculated amount of water and mixing continuously for 10 min. Recipes with different amounts of wild garlic leaves were processed using a single screw extrusion-cooker TS-45 (ZMCh Metalchem, Gliwice, Poland) with L:D = 12:1 (L is barrel length; D is screw diameter). The screw compression ratio was 3:1. The range of the temperatures of the extrusion-cooking process was as follows: 125/130/135 °C, respectively, in the three extruder sections. The extrusion-cooking was carried out at screw speeds: 80 and 120 rpm. A single-open forming die of 3 mm in diameter was used. Samples were stored in polyethylene bags at room temperature before tests [19].

### 3.3. Extraction Procedure

The extraction process was performed in an ultrasonic bath (Bandelin Electronic GmbH & Co. KG, Berlin, Germany). Accordingly, 2 g portions of samples were extracted 2-times with 40 mL of 80% ethanol for 40 min at a temperature of 60 °C, ultrasound frequency of 33 kHz, and a power of 320 W. The extracts were filtered, combined, evaporated to dryness, and dissolved in 10 mL of methanol [19].

### 3.4. Hydrolysis of the Samples

Hydrolysis of the samples was carried out according to the modified method of Czaban et al. [54]. Samples were treated with 2 M NaOH for 4 h at room temperature. At the beginning of hydrolysis, 4 µg of an internal standard (4-OH-benzoic acid) was added to all samples. The hydrolyzed samples were subsequently cooled and acidified using ice-cold 6 M HCl to achieve a pH value of about 2. The resulting mixtures were centrifuged at 8000 rpm for 20 min, and supernatants were extracted with ethyl acetate. The organic phase was collected and evaporated to dryness. The residue was dissolved in methanol and stored in a refrigerator. After centrifugation, the samples were subjected to chromatographic analysis.

### 3.5. Determination of Phenolic Acids

The determination of phenolic acids was carried out according to the modified method of Burda and Oleszek [55]. The phenolic acid content in extracts was assessed by reversed-phase ultra-high pressure liquid chromatography, performed on a Waters ACQUITY UPLC^®^ Systems chromatograph (Waters Corporation, Milford, MA, USA), equipped with a photodiode array detector and coupled to a triple-quadrupole mass spectrometer (Waters ACQUITY^®^ TQD, Micromass, Manchester, GB). Samples were then separated on a Waters ACQUITY UPLC^®^ HSS C18 column (1.0 mm × 100 mm; 1.8 μm) at 30 °C. The mobile phase consisted of 0.1% formic acid in MilliQ water (*v*/*v*) and 0.1% formic acid in acetonitrile (*v*/*v*). The analytes were eluted using a combination of isocratic and gradient steps.

The detection of phenolic acids was performed in the negative ionization mode, using a selected reaction monitoring method. The source temperature was 110 °C, while the desolvation temperature was 350 °C. Nitrogen was used as a desolvation gas (a flow of 1000 L/h) and as a cone gas (100 L/h). Argon was used as a collision gas (0.1 mL/min). Collision energies were optimized for particular phenolic acids. Concentrations of phenolic acids in wheat extracts were calculated on the basis of calibration curves.

### 3.6. Determination of the Total Content of Polyphenolic Compounds (TPC)

The total content of polyphenolic compounds (TPC) was resolved to utilize the modified Folin-Ciocalteu (FC) method [19]. Here, the number of polyphenols is expressed as mg gallic acid equivalents (GAE) per g of dry weight (d.w.).

### 3.7. Ability to Scavenge DPPH

Measurement of antiradical activity was carried out via DPPH stable radical (2,2-diphenyl-1-picrylhydrazyl) spectroscopy, according to the modified method of Burda and Oleszek [55]. Absorbance was measured at 517 nm wavelength every 5 min for 15 min, using a UV-VIS spectrophotometer (Genesys UV-VIS, Thermo Scientific, Waltham, MA, USA). This method enables tracking of absorbance changes over time and indicates the plateau phase. The free radical scavenging ability of the samples was calculated using the following formula:(1)$$\%\mathrm{RSA}=\left[\frac{(A_{0}-A_{1)}}{A_{0}}\right]\times 100$$

A_0_—the absorbance of the sample except for tested extracts

A_1_—the absorbance of the sample with tested extracts

### 3.8. In Vitro Two-Stages Digestion Model

Authors applied a static in vitro digestion model comparing two-stages digestion (gastric and duodenal) according to Seraglio et al. [8] with minor modifications. In order to carry out the first gastric digestion step, 1.632 g of each sample was weighed and homogenized beforehand. To each of the samples, 5.84 mL of gastric solution was added and manually stirred for 4 min. Subsequently, 2.32 mL of hydrochloric acid (Chempur) at pH 2.5 ± 0.2 was added. Afterward, the samples were incubated in a water bath with shaking (GFL 1083) for two hours (37 °C, 100 rpm). The samples were then centrifuged (10 min, 8000 rpm), and the supernatant was collected for further analysis. Until the scheduled analysis was performed, the samples were refrigerated (−20 °C for 24 h).

The next step (duodenal digestion) was performed in the same way as the gastric but using a different amount of sample (2.246 g). After incubation (skipping the centrifugation step), 0.09 mL of 1 mol/L sodium bicarbonate (Chempur) (to increase the pH to 5.5) and 2.26 mL of duodenal solution were added to each sample. Then the samples were stirred for 1 min. 

After this time, 0.72 mL of sodium bicarbonate solution for adjustment to pH 6.7 ± 0.2 was added to each flask. The samples were then incubated in a water bath with shaking for 2 h (37 °C, 100 rpm). After centrifugation (10 min, 8000 rpm), the supernatant was examined and further analyzed. The storage conditions were the same as for the first step.

The simulated gastric juice (gastric solution) was prepared as follows: 0.16 g of pepsin (Merck, Darmstadt, Germany) was dissolved in 0.35 mL of 12 M hydrochloric acid, and it was made up to 50 mL with ultrapure water. Simulated intestinal juice (duodenal solution) was prepared by connecting 0.25 g pancreatin with 0.047 g of sodium glycodeoxycholate, 0.0505 g of sodium taurocholate, and 0.029 g of sodium taurodeoxycholate (Merck) dissolved in 0.25 mL of 0.5 M sodium bicarbonate in 25 mL ultrapure water.

### 3.9. Statistical Analysis

All the measurements were done in three replications; the results were mean values of multiple repetitions and standard deviations (SD). Statistical analysis with ANOVA (Statistica 13.0, StatSoft Inc., Tulsa, OK, USA) was used to determine the significance of differences at α = 0.05, with Duncan’s test applied to evaluate the homogenous groups. Pearson’s correlation coefficients and their significance were evaluated at 0.05 and 0.01 for the tested characteristics.

## 4. Conclusions

Interest in a healthy lifestyle and the prevention of lifestyle diseases have contributed to the advancement of new nutritional trends. This paper describes studies of newly developed innovative snacks enriched with *Allium ursinum* L. leaves. The influence of wild garlic addition and screw rotational speed on the content of polyphenols and biological properties was examined. The highest content of polyphenols and free phenolic acids and the highest radical scavenging activity was found in the fried snacks enriched with 4% of leaves. Polyphenols content and antiradical properties of the snacks exhibited a positive correlation with the addition of wild garlic leaves to the product.

Optimized extrusion-cooking process conditions may release phenolic acids from the chemical bonds that they create with other compounds without deactivating aglycones. The obtained findings demonstrated that for all snacks enriched with the wild garlic, a screw speed of 120 rpm resulted in the highest content of polyphenols and free phenolic acids, as well as a higher ability to scavenge DPPH.

Plant foods have diverse compositions and are often eaten in conjunction with other foods. Therefore, the food bolus ingredients can modulate the bioaccessibility and stability of phytochemicals. Hence, in bioavailability studies, biochemical and chemical reactions and physical barriers occurring within food must be considered. In this study, authors used a two-stage in vitro digestion model, including gastric and duodenal phases. For all the investigated snacks, the concentration of polyphenols after in vitro digestion was significantly reduced, after gastric and duodenal digestion, as compared with the samples before digestion. The content of free phenolic acids also decreased drastically after the first stage of in vitro digestion (gastric), which was deepened during the second stage (duodenal). In addition, the authors found that the in vitro digestion procedure reduced the snack’s antioxidant activity.

This work is the first to comprehensively investigate the in vitro digestion of snacks incorporating polyphenol-rich plant material.

## Figures and Tables

**Figure 1 ijms-23-14458-f001:**
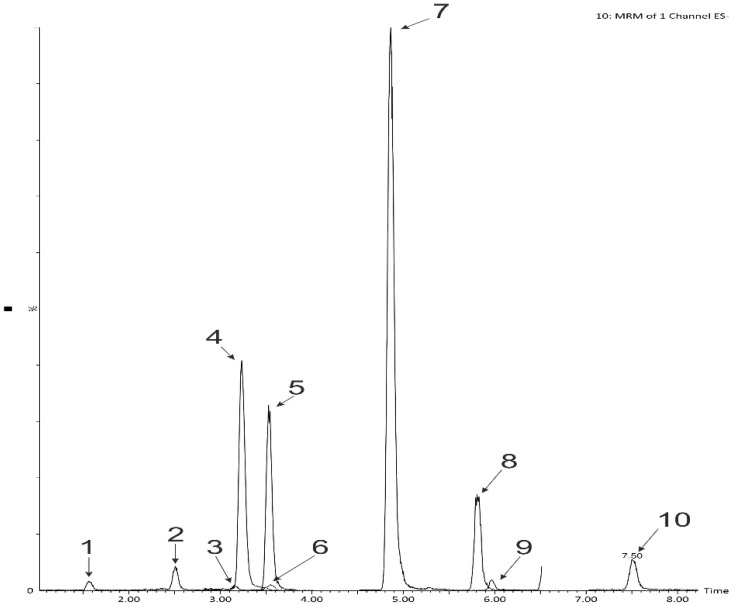
LC-MS-MRM chromatogram of phenolic acids found in snacks enriched with wild garlic (120 rpm, 3% of wild garlic, sample after hydrolysis): 1—protocatechuic, 2—4-OH-benzoic, 3—vanillic, 4—caffeic, 5—3-OH -benzoic (internal standard), 6—syringic, 7—*p*-coumaric, 8—ferulic, 9—sinapic, 10—salicylic.

**Figure 2 ijms-23-14458-f002:**
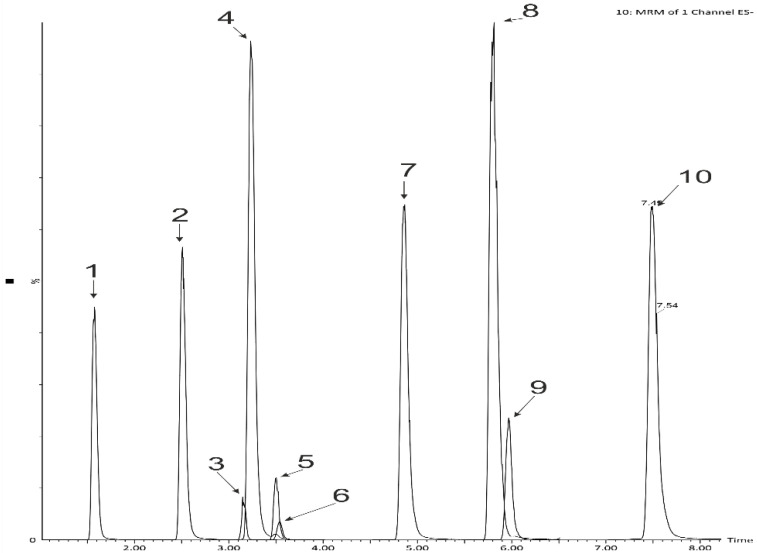
LC-MS-MRM chromatogram of phenolic acids standards: 1—protocatechuic, 2—4-OH-benzoic, 3—vanillic, 4—caffeic, 5—3-OH -benzoic (internal standard), 6—syringic, 7—*p*-coumaric, 8—ferulic, 9—sinapic, 10—salicylic.

**Figure 3 ijms-23-14458-f003:**
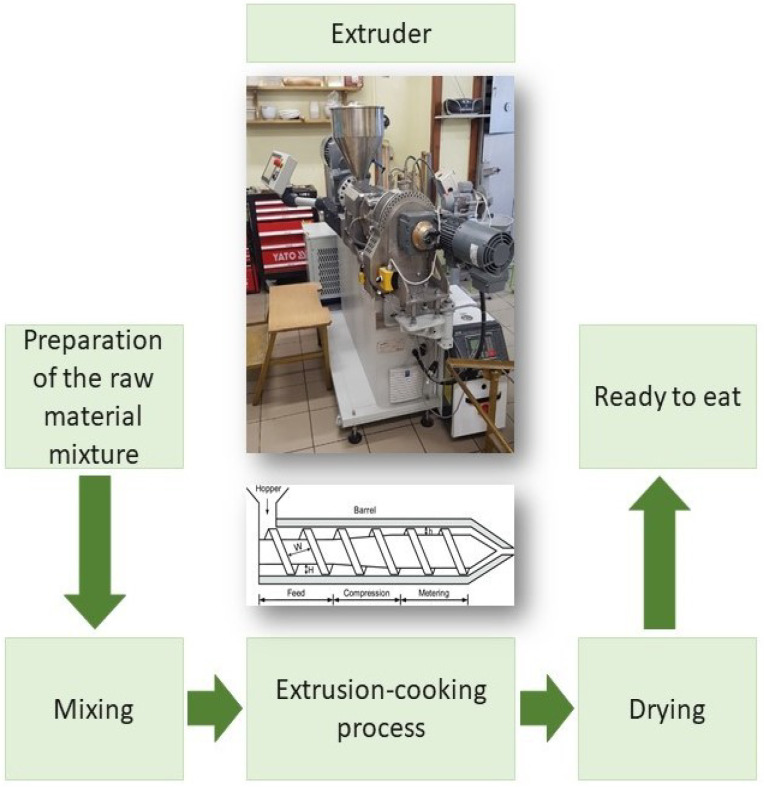
Diagram of the snack production process using extrusion-cooking.

**Table 1 ijms-23-14458-t001:** The total content of polyphenolic compounds (TPC mg GAE/g d.w.) in snacks enriched with wild garlic extruded at two different screw speeds—80 and 120 rpm (n = 3; mean ± SD).

Wild Garlic Addition	Polyphenols Content, 80 rpm	Polyphenols Content, 120 rpm
0%	0.288 ± 0.008	0.381 ± 0.002
1%	0.302 ± 0.007	0.400 ± 0.003
2%	0.388 ± 0.002	0.403 ± 0.022
3%	0.440 ± 0.011	0.445 ± 0.016
4%	0.603 ± 0.009	0.610 ± 0.018

**Table 2 ijms-23-14458-t002:** The content of phenolic acids (after hydrolysis) in snacks enriched with wild garlic extruded at two different screw speeds (n = 3; mean ± SD).

Content of Phenolic Acid (µg/g d.w.)											
Screw Speed	Garlic Addition	Protokatechuic	4-OH-Benzoic	Vanilic	Caffeic	Syringic	Coumaric	Ferulic	Sinapic	Salicylic	Sum
80 rpm	0%	0.406 ± 0.01	4.012 ± 0.20	0.121 ± 0.00	30.358 ± 1.72	8.817 ± 0.21	64.058 ± 2.36	665.751 ± 3.22	63.950 ± 1.15	0.657 ± 0.02	838.130 ± 8.89
	1%	0.574 ± 0.02	3.915 ± 0.16	0.873 ± 0.01	49.192 ± 2.04	11.088 ± 0.54	165.700 ± 5.24	1159.250 ± 12.06	82.192 ± 0.97	0.877 ± 0.03	1473.677 ± 21.07
	2%	0.509 ± 0.02	4.961 ± 0.18	5.083 ± 0.05	63.217 ± 0.39	11.191 ± 0.27	184.117 ± 2.35	1216.350 ± 1.28	93.850 ± 0.89	0.909 ± 0.03	1580.187 ± 5.46
	3%	0.624 ± 0.03	5.026 ± 0.20	21.868 ± 0.19	66.137 ± 1.49	11.253 ± 0.09	196.575 ± 3.28	1260.917 ± 11.98	93.700 ± 2.08	1.065 ± 0.01	1657.165 ± 19.35
	4%	0.442 ± 0.02	5.124 ± 0.01	52,142 ± 2.04	77.700 ± 0.92	12.065 ± 0.02	212.340 ± 0.34	1322.666 ± 7.64	105.342 ± 2.97	1.155 ± 0.02	1788.975 ± 13.98
120 rpm	0%	0.443 ± 0.00	4.928 ± 0.19	0.123 ± 0.01	32.083 ± 1.13	9.521 ± 0.18	66,875 ± 1.14	516.583 ± 7.09	62.108 ± 2.59	0.635 ± 0.01	693.299 ± 12.34
	1%	0.666 ± 0.01	4.955 ± 0.11	2.609 ± 0.02	58.508 ± 2.32	11.098 ± 0.31	173.200 ± 2.54	1161.750 ± 1.59	81.883 ± 1.38	0.779 ± 0.00	1495.448 ± 8.28
	2%	0.656 ± 0.01	5.050 ± 0.09	10.038 ± 0.31	66.479 ± 1.22	11.266 ± 0.16	187.716 ± 1.18	1225.083 ± 12.58	91.633 ± 2.03	0.904 ± 0.00	1598.825 ± 17.58
	3%	0.714 ± 0.00	5.035 ± 0.21	25.296 ± 0.29	67.554 ± 0.38	10.694 ± 0.08	200.208 ± 5.19	1265.333 ± 7.22	89.525 ± 0.50	1.053 ± 0.02	1665.415 ± 13.89
	4%	0.751 ± 0.00	5.365 ± 0.12	96.375 ± 1.15	78.383 ± 3.09	11.413 ± 0.02	216.650 ± 1.94	1323.583 ± 0.93	102.700 ± 4.02	1.130 ± 0.03	1836.350 ± 11.71

**Table 3 ijms-23-14458-t003:** DPPH radical scavenging activity of corn snacks enriched with wild garlic extruded at two different screw speeds (n = 3; mean ± SD).

	Radical Scavenging towards DPPH (%)									
Time (min)	Wild Garlic Addition (%), 80 rpm					Wild Garlic Addition (%), 120 rpm				
	0	1	2	3	4	0	1	2	3	4
0	19.21 ± 0.19	22.58 ± 0.15	24.69 ± 0.12	27.92 ± 0.46	28.95 ± 0.96	24.11 ± 0.44	25.39 ± 0.59	27.31 ± 0.71	28.92 ± 0.96	30.48 ± 0.71
5	60.75 ± 0.43	67.92 ± 1.72	69.95 ± 0.23	74.69 ± 1.29	77.63 ± 0.79	62.28 ± 2.72	69.15 ± 0.81	72.92 ± 0.84	75.97 ± 1.59	79.18 ± 0.16
10	62.32 ± 0.01	69.65 ± 0.98	76.58 ± 1.14	78.19 ± 0.21	83.16 ± 0.81	65.77 ± 1.34	71.53 ± 0.02	77.95 ± 0.93	80.18 ± 2.21	87.68 ± 0.93
15	62.32 ± 0.00	69.65 ± 0.81	76.50 ± 0.56	78.19 ± 0.93	83.16 ± 1.73	65.77 ± 0.20	71.53 ± 1.13	77.95 ± 1.07	80.18 ± 0.93	87.68 ± 0.13

**Table 4 ijms-23-14458-t004:** Pearson’s correlation coefficients for snacks supplemented with wild garlic leaves addition.

	Total Polyphenols	Free Phenolic Acid	DPPH Radical Scavenging Activity
80 rpm			
Wild garlic content	0.952	0.891	0.979
Total polyphenols		0.749	0.903
Free phenolic acids			0.945
120 rpm			
Wild garlic content	0.849	0.873	0.991
Total polyphenols		0.623	0.859
Free phenolic acids			0.887

**Table 5 ijms-23-14458-t005:** The total content of polyphenolic compounds (TPC mg GAE/g d.w.) in snacks enriched with wild garlic before and after two-stage digestion (n = 3; mean ± SD).

Wild Garlic Addition	Polyphenols Content		
	Before Digestion	Gastric Digestion	Duodendal Digestion
0%	0.221 ± 0.005	0.124 ± 0.002	0.042 ± 0.002
2%	0.307 ± 0.003	0.213 ± 0.0227	0.110 ± 0.0227
4%	0.542 ± 0.016	0.390 ± 0.0184	0.189 ± 0.0184

**Table 6 ijms-23-14458-t006:** The content of phenolic acids (without hydrolysis) in snacks enriched with wild garlic extruded at 120 rpm (n = 3; mean ± SD); I- gastric digestion stage, II- duodendal digestion stage.

Content of Phenolic Acid (µg/g d.w.)											
Conditions	Garlic Addition	Protokatechuic	4-OH-benzoic	Vanillic	Caffeic	Syringic	Coumaric	Ferulic	Sinapic	Salicylic	Sum
Before digestion	0%	0.131 ± 0.003	0.235 ± 0.001	0.456 ± 0.002	1.374 ± 0.022	0.196 ± 0.000	4.374 ± 0.051	2.381 ± 0.003	0.324 ± 0.006	0.082 ± 0.000	9.553 ± 0.088
	2%	0.170 ± 0.001	0.272 ± 0.002	0.588 ± 0.008	1.387 ± 0.017	0.271 ± 0.004	6.121 ± 0.009	3.697 ± 0.071	0.362 ± 0.001	0.156 ± 0.004	13.024 ± 0.117
	4%	0.192 ± 0.002	0.296 ± 0.000	0.599 ± 0.010	1.808 ± 0.041	0.336 ± 0.003	7.878 ± 0.085	5.128 ± 0.011	0.556 ± 0.003	0.193 ± 0.002	16.986 ± 0.002
After digestion	0% I	-	-	-	0.079 ± 0.000	-	0.428 ± 0.001	0.236 ± 0.000	-	-	0.743 ± 0.157
	0% II	-	-	-	-	-	0.218 ± 0.000	-	-	-	0.218 ± 0.000
	2% I	-	-	-	0.132 ± 0.001	-	1.356 ± 0.023	0.397 ± 0.003	-	-	1.885 ± 0.027
	2% II	-	-	-	-	-	0.477 ± 0.001	-	-	-	0.477 ± 0.001
	4% I	-	-	-	0.144 ± 0.001	-	1.893 ± 0.003	0.502 ± 0.007	-	-	2.539 ± 0.011
	4% II	-	-	-	-	-	0.754 ± 0.008	-	-	-	0.754 ± 0.008

**Table 7 ijms-23-14458-t007:** DPPH radical scavenging activity of corn snacks enriched with wild garlic before and after two-stage digestion (n = 3; mean ± SD).

	Radical Scavenging (%) before Digestion			Radical Scavenging (%) after Gastric Digestion			Radical Scavenging (%) after Duodendal Digestion		
Time	Wild Garlic Addition (%)								
	0	2	4	0	2	4	0	2	4
0	15.41 ± 0.05	21.08 ± 0.54	25.91 ±1.06	11.21 ± 0.08	21.08 ± 0.54	28.10 ±0.67	10.32 ± 0.34	20.01 ± 1.15	24.91 ±0.39
5	58.73 ± 0.83	62.22 ± 0.49	75.16 ± 1.78	24.07 ± 0.34	32.76 ± 0.33	36.55 ± 1.17	18.07 ± 0.94	29.17 ± 0.13	31.79 ± 0.45
10	59.15 ± 0.24	73.56 ± 2.09	82.16 ± 0.78	38.23 ± 0.86	51.56 ± 0.76	59.98 ± 0.98	29.23 ± 0.89	38.56 ± 1.26	46.98 ± 0.57
15	59.15 ± 0.00	73.56 ± 0.08	82.16 ± 2.73	38.23 ± 0.12	51.56 ± 2.18	59.98 ± 1.11	29.23 ± 1.07	38.56 ± 1.19	46.02 ± 2.29

## Data Availability

Not applicable.

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
