# Peer review of "Effect of the Production Parameters and In Vitro Digestion on the Content of Polyphenolic Compounds, Phenolic Acids, and Antiradical Properties of Innovative Snacks Enriched with Wild Garlic (Allium ursinum L.) Leaves"

_ijms, 2022, doi:10.3390/ijms232214458_

Round 1
Reviewer 1 Report
The authors have given very detailed information about teh wild garlic and its various usefulness in the introduction section , but it will be helpful if the authors will write little more about the digestion of polyphenols and what it happens and why do they need to do this study.
When you are using HPLC for the detemination and quantitation of phenolic compound, please provide the chromatogram obtained for standards as well as your samples for comparison.
what is DPPH?
what are the standards used for HPLC?
Author Response
Dear Reviewer,
The authors would like to thank the Reviewer for their valuable comments which have helped to improve the quality of the manuscript. The article has been corrected in the change tracking system. The authors hope that the revision of the manuscript and our accompanying responses will be sufficient to make our manuscript suitable for publication.
The authors have given very detailed information about teh wild garlic and its various usefulness in the introduction section , but it will be helpful if the authors will write little more about the digestion of polyphenols and what it happens and why do they need to do this study.
The authors completed the Introduction in accordance with the Reviewer's instructions. The added texts about the digestion of polyphenols and about purpose of this research are marked in red.
When you are using HPLC for the detemination and quantitation of phenolic compound, please provide the chromatogram obtained for standards as well as your samples for comparison.
The authors provided the chromatogram obtained for standards (Figure 2) as well as for selected sample (Figure 1).
Quantitative analysis was performed using the LC-MS method with the use of a triple quadrupole type mass spectrometry detector. The Multiple Reaction Monitoring (MRM) method was used here. This is the most sensitive (most efficient) and highly selective mode of the mass spectrometer. In MRM mode mass analyzers operate at maximum capacity. Not all ions of the analyzed analyte reach the detector, because the fragmentation process
usually produces more than one ion fragment. The number of ions reaching the detector is lower than in the case of parent ion observation (lower absolute intensity),
however, thanks to much higher selectivity, it is possible to obtain much higher sensitivity than with a single quadrupole.The increased selectivity also enables the dynamic
range of the analytical method to be increased. The linear response range of the detector in quadrupole systems exceeds 6 orders of magnitude. Due to the fact that the
MRM mode is very selective, it is possible to perform quantitative measurements for signals with an intensity of several dozen counts per second (cps - Count per second).
Under favorable conditions, it is possible to create quantitative methods while maintaining linearity in the range of 6 orders of magnitude. However, the MRM mode has
a serious limitation. We can only register the presence of compounds for which fragmentation reactions have been defined.
If there are other substances in the test sample, we will not get any information about them.
What is DPPH?
DPPH is 2,2-diphenyl-1-picrylhydrazyl radical. The authors explained the abbreviation in the Introduction
What are the standards used for HPLC?
Commercially available phenolic acids purchased from Merck were used as standards.
Reviewer 2 Report
The manuscript described the use of wild ginger as additive in snacks. The manuscript has some flaws, especially result section. Below are my comments,
Why the author choose to use corn snack, not other food?
Has the author compared ultrasonic bath extraction with hot water extraction? As I know, the energy in ultrasonic bath is very dispersed, thus the extraction is not that efficient.
Avoid in using first-person view (I, we, our, etc.) in the manuscript.
Please provide values in the manuscript.
Please capitalize each word for the second row of Table 2
Line 188-189 and Line 192 – The phrase “in the next stage” used twice. Please check.
Line 209-211 – This is not shown in the table?
Section 2.2 does not show any new result. Please move the discussion into previous sections.
Please provide a diagram or figure for extrusion-cooking.
Please make the conclusion shorter.
Author Response
Dear Reviewer,
The authors would like to thank the Reviewer for their valuable comments which have helped to improve the quality of the manuscript. The article has been corrected in the change tracking system. The authors hope that the revision of the manuscript and our accompanying responses will be sufficient to make our manuscript suitable for publication.
The manuscript described the use of wild ginger as additive in snacks. The manuscript has some flaws, especially result section. Below are my comments.
Why the author choose to use corn snack, not other food?
The corn snacks described in this article are a gluten-free product to be consumed between main meals, recommended for people with gluten intolerance. Generally, authors are interested in health-promoting food produced by extrusion. So far, authors have designed and analyzed instant grits, pasta, pellets, snacks enriched with different plant rich in polyphenols. We have published many of the studies (https://doi.org/10.1155/2018/7830546, https://www.mdpi.com/1420-3049/25/19/4538, https://www.mdpi.com/1420-3049/26/5/1245, https://www.mdpi.com/1420-3049/24/14/2623, https://www.mdpi.com/1420-3049/25/4/916, https://www.mdpi.com/1420-3049/24/7/1262) and we have patented several products (in Poland).
The main research goal of the project is to design and analyze a new line of potential functional food products dedicated to people at risk of developing non-communicable chronic diseases.
Enriched food allows obtaining a product desired from the consumer's point of view. Expanding research and knowledge in the field of functional food is, above all, an opportunity to introduce to the market products aimed at consumers with special nutritional needs and susceptible to various diseases.
Taking into account the insufficient prevention of civilization diseases, we develop and research a new range of food products enriched with plant additives rich in biological active substances.
Has the author compared ultrasonic bath extraction with hot water extraction? As I know, the energy in ultrasonic bath is very dispersed, thus the extraction is not that efficient.
Authors would like to thank the Reviewer for the valuable suggestion. In further research, we will try hot water extraction. However, the authors optimized the conditions of ultrasonic extraction (Industrial Crops and Products 83 (2016) 359–363, Food Anal. Methods DOI 10.1007/s12161-016-0489-3. In previously conducted, published studies authors compared the effectiveness of UAE with other methods (e.g. with accelerated solvent extraction - http://dx.doi.org/10.1016/j.arabjc.2016.09.003, microwave assisted extraction, accelerated solvent extraction, Soxhlet extraction, heat reflux extraction - Industrial Crops and Products 76 (2015) 509–514). In most cases, UAE was the most effective method of extracting polyphenolic compounds.
Avoid in using first-person view (I, we, our, etc.) in the manuscript.
The authors removed first-person view from the text.
Please capitalize each word for the second row of Table 2
Table 2 have been corrected.
Line 188-189 and Line 192 – The phrase “in the next stage” used twice. Please check.
The phrase have been checked and corrected.
Line 209-211 – This is not shown in the table?
Thank you for your relevant comment. In lines 209-211 the authors gave imprecise sentences. The presented results concern the research conducted by Sani et al. The phrases have already been corrected.
Section 2.2 does not show any new result. Please move the discussion into previous sections.
Thank you for the comment. The authors removed commonly known information and combined section 2.1 Influence of A. Ursinum Addition on Polyphenols Content, Free Phenolic Acid Content and Antioxidant Properties of Snacks and 2.2 Influence of the Screw Speed on Polyphenols Content, Free Phenolic Acid Content and Antioxidant Properties of Extruded Snacks
Please provide a diagram or figure for extrusion-cooking
Authors provided diagram of manufacturing process of snacks (Figure 3).
Please make the conclusion shorter.
The conclusions have been shortened according to the Reviewer's comment.
Round 2
Reviewer 2 Report
The manuscript can be accepted